# PRUC & Play: Probabilistic Residual User Clustering for Recommender Systems

## Abstract

Modern recommender systems are typically based on deep learning (DL) models, where a dense encoder learns representations of users and items. As a result, these systems often suffer from the black-box nature and computational complexity of the underlying models, making it difficult to systematically interpret their outputs and enhance their recommendation capabilities. To address this problem, we propose *Probabilistic Residual User Clustering (PRUC)*, a causal Bayesian recommendation model based on user clustering. Specifically, we address this problem by (1) dividing users into clusters in an unsupervised manner and identifying causal confounders that influence latent variables, (2) developing sub-models for each confounder given the observable variables, and (3) generating recommendations by aggregating the rating residuals under each confounder using do-calculus. Experiments demonstrate that our *plug-and-play* PRUC is compatible with various base DL recommender systems, significantly improving their performance while automatically discovering meaningful user clusters.

## 1 Introduction

Over the past decade, personalized recommendations have significantly improved user experiences in domains such as e-commerce and social media. The recommender systems driving these advancements often rely on sophisticated deep learning (DL) models (Chung et al., 2014; Vaswani et al., 2017; Wu et al., 2019) capable of handling vast amounts of data, enabling highly accurate predictions and personalized interactions. Despite their effectiveness, these models often function as black boxes, lacking transparency and interpretability. This limitation poses significant challenges, particularly when diagnosing and enhancing the performance of recommender systems in scenarios involving domain shifts, such as changes in users' countries. Cold-start scenarios, a critical problem in recommendation systems, exacerbate these issues due to the presence of heterogeneous features and the influence of diverse and spurious patterns. As a result, existing models exhibit notably low performance in such settings.

Existing work (Yuan et al., 2020; Wu et al., 2020; Bi et al., 2020; Li et al., 2019; Hansen et al., 2020; Liang et al., 2020; Zhu et al., 2020; Liu et al., 2020) often addresses domain shift by establishing connections across different domains through shared users or items. However, in real-world applications, such overlap is often unavailable. For instance, when recommending distinct items to users from different countries, there is typically no overlap in either users or items. This scenario demands more sophisticated modeling to account for shared confounders. For example, consider position/exposure bias in recommender systems: if the system ranks the item (e.g., an ad) higher, users are biased to rate it higher or have a higher probability to click it. Another example is popularity bias; users have a higher probability to click popular or trending items. A system needs to correct for such biases; otherwise the system's accuracy will drop significantly when the once popular items become less popular. Additionally, existing methods often fail to consider latent user clusters when cluster IDs are not available in the datasets, therefore failing to model (dis)similarities among users.

To address these problems, we propose a novel causal hierarchical Bayesian deep learning model, dubbed *Probabilistic Residual User Clustering (PRUC)*, which divides users into latent clusters and makes recommendations based on causal confounders. Our Bayesian causal framework models the residual between the ground-truth rating (or CTR) and the base model's predicted rating, thereby achieving more precise recommendations. Notably, PRUC is *plug-and-play*, meaning that it is compatible with any base DL recommendation model and can enhance the original model's performance.

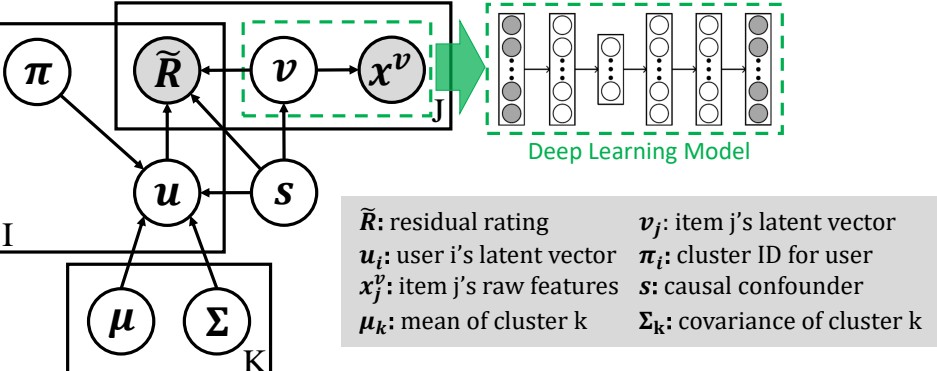

**Figure 1:** Probabilistic graphical model of our PRUC framework.

Our contributions are as follows:

1. We identify the existence of user clusters in various datasets, as well as latent confounders that have a causal effect on user and item hidden representations in DL models.
2. We propose a causal Bayesian framework to discover the latent structures of users, items, and ratings. We incorporate user clusters and causal confounders as latent variables in the causal structural model (SCM) and perform inference via do-calculus over the confounders.
3. We formulate the rating prediction problem as residual prediction, i.e., predicting the difference between the ground-truth user ratings and the base DL model's predicted ratings, to enhance the performance of base DL recommenders.
4. Experiments verify that our *plug-and-play* PRUC is compatible with various base DL recommender systems, significantly improving their performance while automatically discovering meaningful user clusters.

## 2 Probabilistic Residual User Clustering

In this section, we describe our proposed PRUC framework.

### 2.1 Problem Setting and Notations

Consider a recommendation dataset containing $I$ users and $J$ items. A DL encoder $f_v(\cdot) : \mathbb{R}^d \to \mathbb{R}^h$ encodes each item $j$'s raw features $\mathbf{x}_j^v \in \mathbb{R}^d$ into $f_v(\mathbf{x}_j^v)$; assume there exists another decoder deep learning model $f_x(\cdot) : \mathbb{R}^h \to \mathbb{R}^d$, which decodes latent representation $\mathbf{v}_j$ back to the raw item features $\mathbf{x}_j^v$. For a given user $i$ and an item $j$, there is a ground-truth rating $R_{ij} \in \mathbb{R}$, a base predicted rating $\widehat{R}_{ij} \in \mathbb{R}$ provided by a base recommender, and a residual rating $\widetilde{R}_{ij} = R_{ij} - \widehat{R}_{ij}$. There is a latent cluster ID $k$ ($k \in \{1, ..., K\}$) that indicates which user group user $i$ belongs to. We assume that there exists a user latent vector $\mathbf{u}_i \in \mathbb{R}^h$ for each user $i$ and an item latent vector $\mathbf{v}_j \in \mathbb{R}^h$ for each item $j$; they are both impacted by a causal confounder $\mathbf{s} \in \mathbb{R}^g$, where $g \ll h$.

Our goal is to predict the final rating $R$ using the residual $R$, i.e., $R = \widehat{R} + \widetilde{R}$, where $\widehat{R}$ represents the rating from the original (base) DL recommender. When the original recommender is provided, $\widehat{R}$ is fixed; therefore we only need to learn $\widetilde{R}$ in order to predict the final rating $R$. For generality, we assume $M$ domains, with $m_i$ and $m_j$ denoting the domain ID of user $i$ and item $j$, respectively.

### 2.2 Method Overview

We use a variational Bayesian framework to learn the latent parameters. Fig. 1 illustrates the corresponding probabilistic graphical model (PGM).

**Generative Process.** Below we describe the generative process of PRUC shown in Fig. 1.

For each domain $m \in \{1, 2, \dots, M\}$:

- Draw the confounder $\mathbf{s}_m$ from a prior distribution, for example, $p(\mathbf{s}) \sim \mathcal{N}(\mathbf{0}, \mathbf{I})$:

- For each user $i$:
  - Draw the user cluster ID $\pi_i$ from categorical distribution $\pi$.
  - Draw user latent variable $\mathbf{u}_i$ from the $\pi_i$'th Gaussian distribution, i.e., $p(\mathbf{u}_i|\{\boldsymbol{\mu}_k, \boldsymbol{\Sigma}_k\}_{k=1}^K, \mathbf{s}, \pi) \sim \mathcal{N}(\boldsymbol{\mu}_{\pi_i} + \mathbf{W}_u\mathbf{s}_m, \boldsymbol{\Sigma}_{\pi_i})$. Notice that $\mathbf{W}_u$ is the learnable global parameter shared by all users.
  - For each item $j$:
    * Draw item latent variable $v_j$ from distribution $p(\mathbf{v}_j|\mathbf{s}) \sim \mathcal{N}(\mathbf{W}_v\mathbf{s}_m, \lambda_v^{-1}\mathbf{I})$, where $\mathbf{W}_v$ is the learnable global parameter shared by all items, $\mathbf{I}$ is the identity matrix, and $\lambda_v \in \mathbb{R}$ is the precision.
    * Draw the residual rating $\widetilde{R}_{ij}$ from distribution $p(\widetilde{R}_{ij}|\mathbf{u}_i, \mathbf{v}_j, \mathbf{s}) \sim \mathcal{N}(\mathbf{u}_i^\top \mathbf{v}_j + \mathbf{w}_R^\top \mathbf{s}_m, \lambda_{\widetilde{R}_{ij}}^{-1})$, where $\mathbf{w}_R$ is the learnable vector shared by all ratings and $\lambda_{\widetilde{R}_{ij}}$ is the precision.
    * Draw raw item features $\mathbf{x}_j^v$ from distribution $p(\mathbf{x}_j^v|\mathbf{v}_j) \sim \mathcal{N}(f_x(\mathbf{v}_j), \lambda_x^{-1}\mathbf{I})$, where $\mathbf{I}$ is the identity matrix and $\lambda_x \in \mathbb{R}$ is the precision. $f_x$ is a parameterized function that could be learned.

**Model Factorization**. As shown in Fig. 1, we factorize the generative model into five conditional distributions:

$$p(\mathbf{u}_i, \mathbf{v}_j, \mathbf{x}_j^v, \widetilde{R}_{ij}|\{\boldsymbol{\mu}_k, \boldsymbol{\Sigma}_k\}_{k=1}^K, \mathbf{s}_m, \pi) = p(\widetilde{R}_{ij}|\mathbf{u}_i, \mathbf{v}_j, \mathbf{s}_m)p(\mathbf{u}_i|\{\boldsymbol{\mu}_k, \boldsymbol{\Sigma}_k\}_{k=1}^K, \mathbf{s}_m, \pi)p(\mathbf{x}_j^v|\mathbf{v}_j)p(\mathbf{v}_j|\mathbf{s}_m). \quad (1)$$

Each distribution is assumed as a gaussian distribution and is shown as follows:

$$p(\widetilde{R}_{ij}|\mathbf{u}_i, \mathbf{v}_j, \mathbf{s}_m) = \mathcal{N}(\mathbf{u}_i^\top \mathbf{v}_j + \mathbf{w}_R^\top \mathbf{s}_m, \lambda_{\widetilde{R}_{ij}}^{-1}), \quad (2)$$

$$p(\mathbf{u}_i|\{\boldsymbol{\mu}_k, \boldsymbol{\Sigma}_k\}_{k=1}^K, \mathbf{s}_m, \pi) = \mathcal{N}(\boldsymbol{\mu}_{\pi_i} + \mathbf{W}_u\mathbf{s}_m, \boldsymbol{\Sigma}_{\pi_i}), \quad (3)$$

$$p(\mathbf{x}_j^v|\mathbf{v}_j) = \mathcal{N}(f_x(\mathbf{v}_j), \lambda_x^{-1}\mathbf{I}), \quad (4)$$

$$p(\mathbf{v}_j|\mathbf{s}_m) = \mathcal{N}(\mathbf{W}_v\mathbf{s}_m, \lambda_v^{-1}\mathbf{I}), \quad (5)$$

where $i$ and $j$ refers to the user index and the item index, respectively. We employ an inference distribution $q(\mathbf{u}_i, \mathbf{v}_j|\mathbf{x}_j^v)$ to approximate the distribution $p(\mathbf{u}_i, \mathbf{v}_j|\mathbf{x}_j^v)$ for the inference model.

$$q(\mathbf{u}_i, \mathbf{v}_j|\mathbf{x}_j^v) = q(\mathbf{u}_i)q(\mathbf{v}_j|\mathbf{x}_j^v). \quad (6)$$

More specifically, we assumes $q(\mathbf{v}_j|\mathbf{x}_j^v)$ follows a gaussian distribution:

$$q(\mathbf{v}_j|\mathbf{x}_j^v) = \mathcal{N}(f_v(\mathbf{x}_j^v), \Lambda_v^{-1}\mathbf{I}). \quad (7)$$

Here, $j$ is the item index, $\Lambda_v \in \mathbb{R}$ refers to the precision, and $f_v$ is a learnable mapping function.

**Learning Objective.** We maximize an evidence lower bound (ELBO) as our learning objective for both generative and inference model.

$$\mathcal{L}_{ELBO}(\mathbf{x}_j^v, \widetilde{R}_{ij}) = \mathbb{E}_{q(\mathbf{u}_i, \mathbf{v}_j|\mathbf{x}_j^v)}\big[\log p(\mathbf{u}_i, \mathbf{v}_j, \mathbf{x}_j^v, \widetilde{R}_{ij}|\{\boldsymbol{\mu}_k, \boldsymbol{\Sigma}_k\}_{k=1}^K, \mathbf{s}_m, \pi)\big] - \mathbb{E}_{q(\mathbf{u}_i, \mathbf{v}_j|\mathbf{x}_j^v)}\big[\log q(\mathbf{v}_j|\mathbf{x}_j^v)\big]. \quad (8)$$

Combining Eqn. 1 and Eqn. 6, we obtain the following decomposition:

$$\mathcal{L}_{ELBO}(\mathbf{x}_j^v, \widetilde{R}_{ij}) = \mathbb{E}_{q(\mathbf{u}_i)}\big[\log p(\mathbf{u}_i|\{\boldsymbol{\mu}_k, \boldsymbol{\Sigma}_k\}_{k=1}^K, \mathbf{s}_m, \pi)\big] \quad (9)$$

$$+ \mathbb{E}_{q(\mathbf{v}_j|\mathbf{x}_j^v)}\big[\log p(\mathbf{x}_j^v|\mathbf{v}_j)\big] \quad (10)$$

$$+ \mathbb{E}_{q(\mathbf{u}_i, \mathbf{v}_j|\mathbf{x}_j^v)}\big[\log p(\widetilde{R}_{ij}|\mathbf{u}_i, \mathbf{v}_j, \mathbf{s}_m)\big] \quad (11)$$

$$- D_{KL}\big(q(\mathbf{v}_j|\mathbf{x}_j^v)\|p(\mathbf{v}_j|\mathbf{s}_m)\big), \quad (12)$$

where $D_{KL}(\cdot\|\cdot)$ is the Kullback-Leibler (KL) divergence. For Eqn. 9, we compute the log likelihood for each cluster $k$ as

$$\log p(\mathbf{u}_i|\{\boldsymbol{\mu}_k, \boldsymbol{\Sigma}_k\}, \mathbf{s}_m, \pi) = -\frac{1}{2}\sum_{i \in I_k}\big[\log|\boldsymbol{\Sigma}_k| + (\mathbf{u}_i - \boldsymbol{\mu}_k - \mathbf{W}_u\mathbf{s}_m)^\top \boldsymbol{\Sigma}_k^{-1}(\mathbf{u}_i - \boldsymbol{\mu}_k - \mathbf{W}_u\mathbf{s}_m)\big] + C, \quad (13)$$

where $i$ is the user index, $\mathbf{I}_k$ is the set of user index that belongs to cluster $k$, and $C$ is a constant.

Similarly, all the other terms can be expanded as:

$$\log p(\mathbf{x}_j^v|\mathbf{v}_j) = -\frac{\lambda_x}{2}\|\mathbf{x}_j^v - f_x(\mathbf{v}_j)\|^2 + C, \tag{14}$$

$$\log p(\widetilde{R}_{ij}|\mathbf{u}_i, \mathbf{v}_j, \mathbf{s}) = -\frac{\lambda_{\widetilde{R}_{ij}}}{2}\left(\widetilde{R}_{ij} - \mathbf{u}_i^\top \mathbf{v}_j - \mathbf{w}_R^\top \mathbf{s}_m\right)^2 + C, \tag{15}$$

$$D_{KL}\big(q(\mathbf{v}_j|\mathbf{x}_j^v)\|p(\mathbf{v}_j|\mathbf{s}_m)\big) = \frac{\lambda_v}{2}\|\mathbf{v}_j - \mathbf{W}_v\mathbf{s}_m\|^2 - \frac{\Lambda_v}{2}\|\mathbf{v}_j - f_v(\mathbf{x}_j^v)\|^2 + C. \tag{16}$$

**Intuition for Each Term in Eqn. 8.** Below, we describe the intuition of each term in Eqn. 8:

1. **Regularize Latent Variable $\mathbf{u}_i$ (Eqn. 9).** $\mathbb{E}_{q(\mathbf{u}_i)}[p(\mathbf{u}_i|\{\boldsymbol{\mu}_k, \boldsymbol{\Sigma}_k\}_{k=1}^K, \mathbf{s}_m, \pi)]$ aims to regularize user $i$'s latent variable $\mathbf{u}_i$, ensuring $\mathbf{u}_i$ is close to the center of its corresponding user cluster $\pi_i$, and therefore close to other users' latent embeddings in the same cluster.

2. **Reconstruct Data $\mathbf{x}_j^v$ from $\mathbf{v}_j$ (Eqn. 10).** $q(\mathbf{v}_j|\mathbf{x}_j^v)$ and $p(\mathbf{x}_j^v|\mathbf{v}_j)$ are to reconstruct data $\mathbf{x}_j^v$ from the inferred $\mathbf{v}_j$, which encourage the latent variable $\mathbf{v}_j$ to maintain as much relevant information as possible from the raw features $\mathbf{x}_j^v$.

3. **Predict Residual Rating $\widetilde{R}_{ij}$ from $\mathbf{u}_i$ and $\mathbf{v}_j$ (Eqn. 11).** $p(\widetilde{R}_{ij}|\mathbf{u}_i, \mathbf{v}_j, \mathbf{s}_m)$ use the inferred $\mathbf{u}_i$, $\mathbf{v}_j$, and the causal confounder $\mathbf{s}_m$ to predict the residual rating, thereby encouraging $\mathbf{u}_i$ and $\mathbf{v}_j$ to retain more information to maximize prediction performance.

4. **Regularize Latent Variable $\mathbf{v}_j$ (Eqn. 12).** $D_{KL}(q(\mathbf{v}_j|\mathbf{x}_j^v)\|p(\mathbf{v}_j|\mathbf{s}_m))$ is the KL divergence term between the inference model $q(\cdot|\mathbf{x}_j^v)$ and the generative model $p(\cdot|\mathbf{s}_m)$; this encourages the inferred posterior $q(\mathbf{v}_j|\mathbf{x}_j^v)$ to be close to the prior distribution $p(\mathbf{v}_j|\mathbf{s}_m)$.

## 2.3 Inference and Learning

In our framework, we need to learn several parameters, including the Gaussian parameters $\{\boldsymbol{\mu}_k, \boldsymbol{\Sigma}_k\}_{k=1}^K$, user latent $\mathbf{u}$, item latent $\mathbf{v}$, and the parameters of the functions $f_x(\cdot)$ and $f_v(\cdot)$, as well as $\mathbf{W}_u$, $\mathbf{W}_v$, and $\mathbf{w}_R$. The following sections detail the learning process for all these parameters. The complete algorithm is outlined in Algorithm 1.

**1)** $\{\boldsymbol{\mu}_k, \boldsymbol{\Sigma}_k\}_{k=1}^K$. To optimize $\{\boldsymbol{\mu}_k, \boldsymbol{\Sigma}_k\}_{k=1}^K$, we take derivative of Eqn. 13 w.r.t. $\boldsymbol{\mu}_k$ and $\boldsymbol{\Sigma}_k$ as follows:

$$\frac{\partial\mathcal{L}}{\partial\boldsymbol{\mu}_k} = \boldsymbol{\Sigma}_k^{-1}\left(\mathbf{u}_i - \boldsymbol{\mu}_k - \mathbf{W}_u\mathbf{s}_m\right), \tag{17}$$

$$\frac{\partial\mathcal{L}}{\partial\boldsymbol{\Sigma}_k} = \frac{1}{2}\boldsymbol{\Sigma}_k^{-1}\left[\left(\mathbf{u}_i - \boldsymbol{\mu}_k - \mathbf{W}_u\mathbf{s}_m\right)\left(\mathbf{u}_i - \boldsymbol{\mu}_k - \mathbf{W}_u\mathbf{s}_m\right)^\top - \boldsymbol{\Sigma}_k\right]\boldsymbol{\Sigma}_k^{-1}. \tag{18}$$

Setting Eqn. 17 and Eqn. 18 to zero leads to the following update rules, respectively:

$$\boldsymbol{\mu}_k = \frac{1}{|I_k|}\sum_{i\in I_k}\left(\mathbf{u}_i - \mathbf{W}_u\mathbf{s}_m\right), \tag{19}$$

$$\boldsymbol{\Sigma}_k = \frac{1}{|I_k|}\sum_{i\in I_k}\left(\mathbf{u}_i - \boldsymbol{\mu}_k - \mathbf{W}_u\mathbf{s}_m\right)\left(\mathbf{u}_i - \boldsymbol{\mu}_k - \mathbf{W}_u\mathbf{s}_m\right)^\top, \tag{20}$$

where $\mathbf{I}_k$ is the set of user index $i$ that belongs to cluster $k$.

**2)** $\mathbf{u}_i, \mathbf{v}_j$. After computing the gradients of Eqn. 8 w.r.t. to $\mathbf{u}_i$ and $\mathbf{v}_j$, we obtain the following update rules:

$$\mathbf{u}_i = (\boldsymbol{\Sigma}_{\pi_i}\mathbf{V}\lambda_{\widetilde{R}_{(i,:)}}\mathbf{V}^\top + \mathbf{I})^{-1}[\boldsymbol{\mu}_{\pi_i} + \mathbf{W}_u\mathbf{s}_m + \boldsymbol{\Sigma}_{\pi_i}\mathbf{V}\lambda_{\widetilde{R}_{(i,:)}}(\widetilde{\mathbf{R}}_{(i,:)} - \mathbf{w}_R^\top\mathbf{s}_m\mathbf{I})], \tag{21}$$

$$\mathbf{v}_j = [\mathbf{U}\lambda_{\widetilde{R}_{(:,j)}}\mathbf{U}^\top + (\lambda_v - \Lambda_v)\mathbf{I}]^{-1}[\lambda_v\mathbf{W}_v\mathbf{s}_m - \Lambda_v f_v(\mathbf{x}_j^v) + \mathbf{U}\lambda_{\widetilde{R}_{(:,j)}}(\widetilde{\mathbf{R}}_{(:,j)} - \mathbf{w}_R^\top\mathbf{s}_m\mathbf{I})]. \tag{22}$$

Note that here $\mathbf{U}$ and $\mathbf{V}$ refer to user latent matrix $(\mathbf{u}_i)_{i=1}^I$ and item latent matrix $(\mathbf{v}_j)_{j=1}^J$. $\widetilde{\mathbf{R}}_{(i,:)} = (\widetilde{R}_{i1}, \cdots, \widetilde{R}_{iJ})^\top$, $\widetilde{\mathbf{R}}_{(:,j)} = (\widetilde{R}_{1j}, \cdots, \widetilde{R}_{Ij})^\top$. $\lambda_{\widetilde{R}_{(i,:)}} = \mathrm{diag}(\lambda_{\widetilde{R}_{i1}}, \cdots, \lambda_{\widetilde{R}_{iJ}})$, $\lambda_{\widetilde{R}_{(:,j)}} = \mathrm{diag}(\lambda_{\widetilde{R}_{1j}}, \cdots, \lambda_{\widetilde{R}_{Ij}})$.

---

**Algorithm 1** Inference and Learning Algorithm of PRUC

---

**Input:** Raw item features $\mathbf{x}^v$, initialized $f_x(\cdot)$ and $f_v(\cdot)$ parameters, $\mathbf{W}_u, \mathbf{W}_v, \mathbf{w}_R$, initialized Gaussian parameters $\{\boldsymbol{\mu}_k, \boldsymbol{\Sigma}_k\}_{k=1}^K$, and the number of epochs T.
**for** $t = 1 : T$ **do**
  **for** $m = 1 : M$ **do**
    Update $\mathbf{u}_i$ and $\mathbf{v}_j$ using Eqn. 21 and Eqn. 22.
    Update $\mathbf{W}_u, \mathbf{W}_v, \mathbf{w}_R$ using Eqn. 23, Eqn. 24 and Eqn. 25.
    Update the parameters of $f_v(\cdot)$ using gradient ascent of $\mathcal{L}$ in Eqn. 8.
  Update $\{\boldsymbol{\mu}_k, \boldsymbol{\Sigma}_k\}_{k=1}^K$ using Eqn. 19 and Eqn. 20, respectively; update parameters of $f_x(\cdot)$ using gradient ascent of $\mathcal{L}$ in Eqn. 8.
**Output:** $f_x(\cdot)$ and $f_v(\cdot)$ parameters, $\mathbf{W}_u, \mathbf{W}_v, \mathbf{w}_R$, and Gaussian parameters $\{\boldsymbol{\mu}_k, \boldsymbol{\Sigma}_k\}_{k=1}^K$.

---

**3) $\mathbf{W}_u, \mathbf{W}_v, \mathbf{w}_R$.** The update rules for $\mathbf{W}_u, \mathbf{W}_v$, and $\mathbf{w}_R$ are as follows:

$$\mathbf{W}_u = \frac{1}{I}(\sum_{i=1}^{I} \mathbf{u}_i - \sum_{k=1}^{K} |I_k|\boldsymbol{\mu}_k)\mathbf{s}_m^\top(\mathbf{s}_m\mathbf{s}_m^\top)^{-1}, \tag{23}$$

$$\mathbf{W}_v = \frac{1}{J}\sum_{j=1}^{J} \mathbf{v}_j\mathbf{s}_m^\top(\mathbf{s}_m\mathbf{s}_m^\top)^{-1}, \tag{24}$$

$$\mathbf{w}_R = \frac{\sum_{i,j} \lambda_{\widetilde{R}_{ij}}(\widetilde{R}_{ij} - \mathbf{u}_i^\top\mathbf{v}_j)}{\sum_{i,j} \lambda_{\widetilde{R}_{ij}}}(\mathbf{s}_m\mathbf{s}_m^\top)^{-1}\mathbf{s}_m. \tag{25}$$

**4) Parameters of $f_x(\cdot)$ and $f_v(\cdot)$.** We use gradient ascent of $\mathcal{L}$ in Eqn. 8 to update these parameters.

**Inference.** Inference includes the *E-Step* in Algorithm 1, where PRUC updates learnable parameters $\mathbf{W}_u, \mathbf{W}_v, \mathbf{w}_R$, and the parameters of encoder model $f_v(\cdot)$ using gradient ascent of $\mathcal{L}$ in Eqn. 8.

**Learning.** Learning includes the iteration between the *E-Step* and *M-Step* in Algorithm 1 until convergence. In each *M-Step*, we update the Gaussian parameters $\{\boldsymbol{\mu}_k, \boldsymbol{\Sigma}_k\}_{k=1}^K$ following the update rule from Eqn. 19 and Eqn. 20, respectively; we also update parameters of decoder model $f_x(\cdot)$ using gradient ascent of $\mathcal{L}$ in Eqn. 8.

## 2.4 Plug-and-Play PRUC

Below we discuss the key components of our plug-and-play PRUC as a Bayesian causal inference framework.

**Inferring User Cluster $\pi_i$.** With the learned Gaussian mixture's parameters, i.e., the mean and covariance $\boldsymbol{\mu}_k$ and $\boldsymbol{\Sigma}_k$ for each Gaussian component $k$ (each *Gaussian component* represents one *user cluster*), PRUC infers the cluster for each user $i$, i.e., $p(\pi_i|\widetilde{R}_{ij}, \{\mathbf{u}_i\}, \{\mathbf{v}_j\}, \{\boldsymbol{\mu}_k, \boldsymbol{\Sigma}_k\}_{k=1}^K)$, i.e., which cluster $\pi_i$ user $i$ belongs to.

**Isolating Causal Confounders $\mathbf{s}_m$.** With the learned structured causal model (SCM), we isolate the *causal confounders* $\mathbf{s}_m$ for each domain $m$ by approximating its posterior distribution $p(\mathbf{s}_m|\widetilde{R}, \mathbf{x}_j^v, \{\boldsymbol{\mu}_k, \boldsymbol{\Sigma}_k\}_{k=1}^K)$ via variational domain indexing (VDI) (Xu et al., 2023). In this way, we can minimize the bias introduced by the causal confounder $\mathbf{s}_m$ when inferring $\mathbf{u}_i$ and $\mathbf{v}_j$ using Eqn. 3 and Eqn. 7, respectively.

**Debiasing the Causal Confounders.** Under our *PRUC* framework, for each inferred user cluster $k$, we perform causal inference for each user $i$ in the cluster to predict the residual $\widetilde{R}_{ij}$ (for each item $j$) while debiasing the causal confounders $\mathbf{s}$. Specifically, with inferred $\mathbf{u}_i$ and $\mathbf{v}_j$, we can predict $\widetilde{R}_{ij}$ by do-calculus as

$$p^{(k)}(\widetilde{R}_{ij}|do(\mathbf{u}_i), do(\mathbf{v}_j)) = \sum_{m=1}^{M} p^{(k)}(\widetilde{R}_{ij}|\mathbf{u}_i, \mathbf{v}_j, \mathbf{s}_m)p(\mathbf{s}_m), \tag{26}$$

where $p^{(k)}(\widetilde{R}_{ij}|\mathbf{u}_i, \mathbf{v}_j, \mathbf{s})$ represents the $k$'th sub-model trained from the $k$'th cluster's user data. In practice, we use $k = \pi_i$ ($\pi_i$ is user $i$'s cluster) when predicting user $i$'s rating $\widetilde{R}_{ij}$.

Note that performing causal inference by intervening $(\mathbf{u}_i, \mathbf{v}_j)$ effectively cuts the relations between the causal confounders $\mathbf{s}$ and $(\mathbf{u}_i, \mathbf{v}_j)$. Fig. 2 demonstrate the do-calculus that PRUC performs for debiasing the causal confounder $\mathbf{s}$.

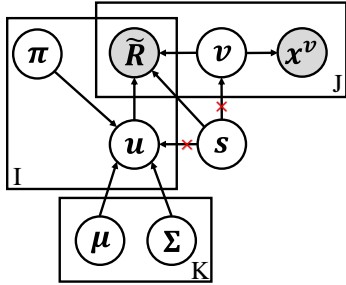

**Intuition behind Do-Calculus.** The rationale of performing do-calculus in PRUC is that getting interventional distributions often requires intervening the recommender system to collect training data, which is expensive in practice. In contrast, do-calculus works by leveraging existing data to estimate the conditional distribution $p^{(k)}(\widetilde{R}_{ij}|\mathbf{u}_i, \mathbf{v}_j, \mathbf{s})$, and therefore prevent the potential cost (and risk) of actually intervening the system.

**Figure 2:** Causal inference in PRUC is equivalent to cutting the the confounder $\mathbf{s}$'s influence on $\mathbf{u}_i$ and $\mathbf{v}$.

**Summary.** To summarize, for each user $i$, PRUC causally infer the residual rating $\widetilde{R}_i$ as follows:

1. Infer the user cluster $\pi_i$ by approximating its posterior $p(\pi_i|\mathbf{u}_i, \mathbf{v}_j, \mathbf{x}_j^v, \{\boldsymbol{\mu}_k, \boldsymbol{\Sigma}_k\}_{k=1}^K)$.
2. Infer the residual rating $\widetilde{R}_{ij}$ by causal Bayesian model averaging defined in Eqn. 26.
3. Predict the final rating as $R = \widetilde{R} + \widehat{R}$, where $\widehat{R}$ is the base recommender's prediction.

## 3 Experiments

In this section, we evaluate our PRUC as a plug-and-play framework to enhance arbitrary base recommenders on *XMRec*, which contains data from 18 countries.

### 3.1 Datasets

*XMRec* (Bonab et al., 2021) is a collection of datasets that encompass 18 local markets (i.e., countries), 16 distinct product categories, and 52.5 million user-item interactions. For each item $j$, we use its item descriptions from the dataset as the item features $\mathbf{x}_j^v$. To minimize unnecessary noise, users who have made fewer than three purchases are excluded from our experiments

**Table 1:** Three source-target domain splits for XMRec.

| Split | Source Domains | Target Domains |
|---|---|---|
| 1 | France, Italy, India | Japan, Mexico |
| 2 | Mexico, Spain, India | Japan, Germany |
| 3 | Germany, Italy, Japan | United States, India |

for all models. Table 1 shows the source-target domain splits for XMRec. For example, in Split 1, we use France, Italy, and India as the source domains and Japan and Mexico as the target domains. The goal is to improve performance in the target domains.

In all experiments, we focus on the cold-start setting where for the target domains, only one rating per user is available in the training set, making the recommendation task extremely challenging.

### 3.2 Base Recommenders and Baselines

Note that our PRUC method is a *plug-and-play* solution, compatible with *any* base recommenders. In this paper, we select the following three base recommenders as base models to demonstrate that PRUC can enhance state-of-the-art recommendation models.

- **CDL** (Wang et al., 2015) is a hierarchical Bayesian framework that jointly integrates deep representation learning of content information with collaborative filtering on the ratings (feedback) matrix within a unified model.
- **DLRM** (Naumov et al., 2019) is a deep learning recommendation model that uses embeddings to represent sparse and dense features and predicts event probability.
- **PerK** (Kweon et al., 2024) is a recommendation approach that calculates the expected user utility by leveraging calibrated interaction probabilities and selects the recommendation size $K$ that maximizes this utility.

Here CDL, DLRM, and PerK serve as both (1) our **baselines** to compare against and (2) our **base recommenders** to enhance (see Fig. 1). Our experiments below will show that our PRUC can be

plugged in to any of these base recommenders and improve their recommendation performance. For more details on training configurations, see Appendix A.2.

## 3.3 Metrics

We use five metrics for evaluation: Recall, NDCG, F1, Precision, and mAP.

**Recall.** Recall@$N$ measures the proportion of relevant items retrieved among the top $N$ recommended items for user $i$. It is defined as:

$$\text{Recall}_i@N = \frac{\sum_{n=1}^{N} \text{rel}_{i,n}}{|J_i|}, \tag{27}$$

where $\text{rel}_{i,n}$ is an indicator that equals 1 if the item at rank $n$ is relevant to user $i$, and 0 otherwise. $|J_i|$ denotes the total number of relevant items for user $i$.

**Precision.** Precision@$N$ measures the proportion of the top $N$ recommended items that are relevant to user $i$. It is defined as:

$$\text{Precision}_i@N = \frac{\sum_{n=1}^{N} \text{rel}_{i,n}}{N}, \tag{28}$$

where $\text{rel}_{i,n}$ is 1 if the item at rank $n$ is relevant to user $i$, and 0 otherwise.

**mAP.** Mean Average Precision (mAP) computes the average precision over all relevant items for user $i$. See Appendix A.1 for more details.

**F1-score.** The F1 Score@$N$ for user $i$ is the harmonic mean of Precision@$N$ and Recall@$N$, providing a balance between the two metrics:

$$\text{F1}_i@N = 2 \times \frac{\text{Precision}_i@N \times \text{Recall}_i@N}{\text{Precision}_i@N + \text{Recall}_i@N}, \tag{29}$$

where $\text{Precision}_i@N$ and $\text{Recall}_i@N$ are as previously defined for user $i$ at rank $N$.

**NDCG.** Normalized Discounted Cumulative Gain (NDCG@$N$) evaluates the quality of the ranked list by considering the positions of the relevant items, giving higher scores to items appearing earlier in the list. See Appendix A.1 for more details.

Note that all metrics are computed by averaging over all users $i$.

## 3.4 Results

**Results for Different Base Models.** Table 2 shows the performance of our PRUC with different base models, i.e., CDL, DLRM, and PerK in terms of different metrics. We can see that our PRUC, even without the causality component (i.e., "PRUC w/o Causality") can often enhance the performance of different base models, and our full PRUC (i.e., "PRUC (Full)") can further improve the recommendation performance.

**Recall@$N$ with Larger $N$.** Fig. 3 shows Recall@$N$ for $N = 50, 100, 150, 200, 250, 300$ across all three base models (CDL, DLRM, and PerK) and three source-target domain splits (Table 1). These figures indicate that PRUC, even without causality, consistently outperforms the base models, while our full PRUC consistently outperforms PRUC without causality in all settings.

**Results for Different Clusters Discovered by PRUC.** Table 3, Table 4, and Table 5 show the performance of our PRUC with CDL, DLRM, and PerK as the base model (base recommender). We can see that our PRUC, even without the causality component (i.e., "PRUC w/o Causality") can often enhance the performance of the base model consistently across all clusters. Besides, our full PRUC (i.e., "PRUC (Full)") can further improve the recommendation performance. For example, CDL as the base model achieves a recall@20 of 0.0241 for User Cluster 1 in the domain split of "France, Italy, India $\to$ Japan, Mexico". Our PRUC without the causal inference component improves the recall to 0.0278. Our full PRUC then further improves its recall@20 to 0.0708.

**Visualization of PRUC's Discovered User Clusters.** Fig. 4 shows the visualization of the discovered user clusters from PRUC with base models CDL (**left**), DLRM (**middle**), and PerK (**right**) for the domain split "France, Italy, India $\to$ Japan, Mexico".

**Table 2:** Performance of PRUC with different base models. The best results are marked with **bold face**.

| Data | Method | Recall@20 | F1@20 | MAP@20 | NDCG@20 | Precision@20 |
|---|---|---|---|---|---|---|
| France, Italy, India →Japan, Mexico | CDL (Base Model) | 0.0143 | 0.0016 | 0.0028 | 0.0009 | 0.0009 |
| | PRUC w/o Causality | 0.1058 | 0.0126 | 0.0333 | 0.0088 | 0.0067 |
| | PRUC (Full) | **0.1091** | **0.0128** | **0.0463** | **0.0108** | **0.0068** |
| | DLRM (Base Model) | 0.0044 | 0.0004 | 0.0004 | 0.0002 | 0.0002 |
| | PRUC w/o Causality | 0.0232 | 0.0026 | 0.0039 | 0.0014 | 0.0014 |
| | PRUC (Full) | **0.0295** | **0.0035** | **0.0048** | **0.0018** | **0.0018** |
| | PerK (Base Model) | 0.1098 | 0.0128 | 0.0512 | 0.0112 | 0.0068 |
| | PRUC w/o Causality | 0.1376 | 0.0160 | 0.0558 | 0.0129 | 0.0085 |
| | PRUC (Full) | **0.1634** | **0.0189** | **0.0626** | **0.0148** | **0.0100** |
| Mexico, Spain, India →Japan, Germany | CDL (Base Model) | 0.1127 | 0.0135 | 0.0301 | 0.0086 | 0.0072 |
| | PRUC w/o Causality | 0.1688 | 0.0209 | 0.0573 | 0.0151 | 0.0111 |
| | PRUC (Full) | **0.1837** | **0.0238** | **0.0605** | **0.0168** | **0.0127** |
| | DLRM (Base Model) | 0.0756 | 0.0093 | 0.0085 | 0.0041 | 0.0049 |
| | PRUC w/o Causality | 0.1455 | 0.0181 | 0.0275 | 0.0098 | 0.0097 |
| | PRUC (Full) | **0.2017** | **0.0246** | **0.0545** | **0.0156** | **0.0131** |
| | PerK (Base Model) | 0.1443 | 0.0177 | 0.0601 | 0.0143 | 0.0094 |
| | PRUC w/o Causality | 0.2260 | 0.0269 | 0.0920 | 0.0219 | 0.0142 |
| | PRUC (Full) | **0.2641** | **0.0322** | **0.1082** | **0.0258** | **0.0171** |
| Germany, Italy, Japan →United States, India | CDL (Base Model) | **0.0252** | **0.0055** | **0.0084** | **0.0040** | **0.0031** |
| | PRUC w/o Causality | 0.0194 | 0.0045 | 0.0049 | 0.0030 | 0.0026 |
| | PRUC (Full) | 0.0222 | 0.0053 | 0.0066 | 0.0037 | 0.0030 |
| | DLRM (Base Model) | 0.0024 | 0.0006 | 0.0003 | 0.0003 | 0.0003 |
| | PRUC w/o Causality | 0.0045 | 0.0012 | 0.0011 | 0.0008 | 0.0007 |
| | PRUC (Full) | **0.0066** | **0.0016** | **0.0024** | **0.0012** | **0.0009** |
| | PerK (Base Model) | 0.0148 | 0.0033 | 0.0041 | 0.0022 | 0.0018 |
| | PRUC w/o Causality | 0.0197 | 0.0044 | 0.0054 | 0.0029 | 0.0025 |
| | PRUC (Full) | **0.0206** | **0.0046** | **0.0059** | **0.0031** | **0.0026** |

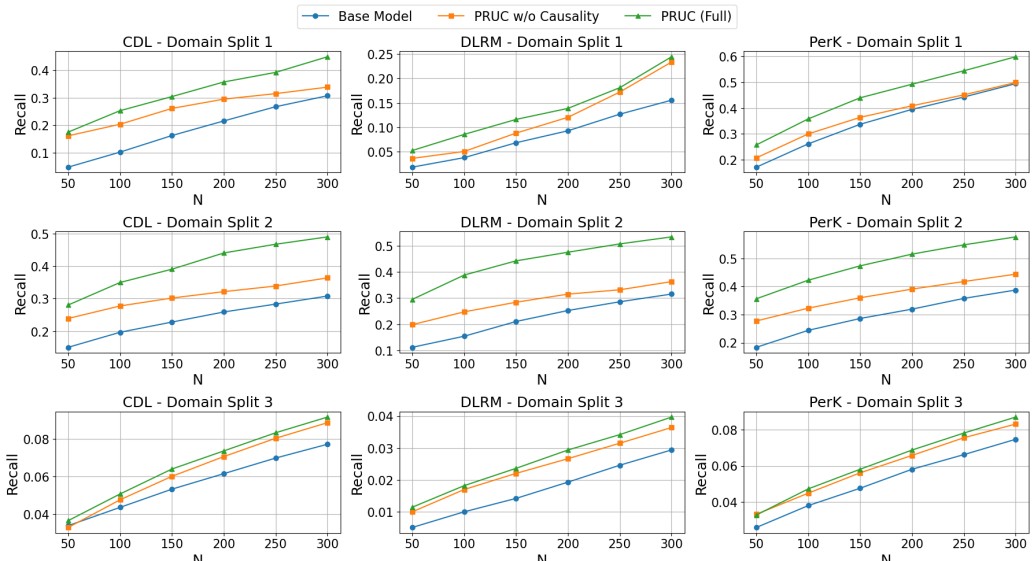

**Figure 3:** Recall@$N$ on XMRec for PRUC with three base models, i.e., CDL, DLRM, and PerK.

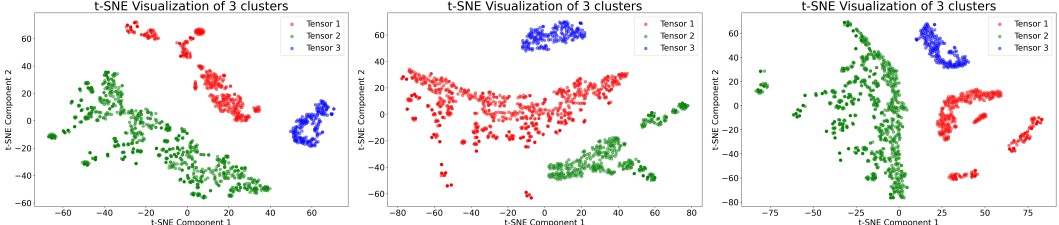

**Figure 4:** Visualization of the discovered user clusters from PRUC with base models CDL (**left**), DLRM (**middle**), and PerK (**right**) for the domain split "France, Italy, India → Japan, Mexico".

**Ablation Study.** Comparing the performance of "PRUC w/o Causality" and "PRUC (Full)" in both Table 2 and Fig. 3, we can see that the full PRUC often outperforms the original "PRUC w/o Causal-

**Table 3:** Performance of PRUC on different user clusters with CDL as the base model. "-" means a cluster contains only training-set users, i.e., no test-set users to evaluate. The best results are marked with **bold face**.

| Data | Cluster | Method | Recall@20 | F1@20 | MAP@20 | NDCG@20 | Precision@20 |
|---|---|---|---|---|---|---|---|
| France, Italy, India →Japan, Mexico | 1 | CDL (Base Model) | 0.0241 | 0.0028 | 0.0062 | 0.0018 | 0.0015 |
| | | PRUC w/o Causality | 0.0278 | 0.0033 | 0.0056 | 0.0018 | 0.0017 |
| | | PRUC (Full) | **0.0708** | **0.0074** | **0.0652** | **0.0105** | **0.0039** |
| | 2 | CDL (Base Model) | 0.0126 | 0.0014 | 0.0022 | 0.0007 | 0.0008 |
| | | PRUC w/o Causality | 0.0075 | 0.0007 | 0.0007 | 0.0003 | 0.0004 |
| | | PRUC (Full) | **0.1156** | **0.0138** | **0.0431** | **0.0109** | **0.0073** |
| | 3 | CDL (Base Model) | - | - | - | - | - |
| | | PRUC w/o Causality | 0.1720 | 0.0205 | 0.0564 | 0.0146 | 0.0109 |
| | | PRUC (Full) | - | - | - | - | - |
| Mexico, Spain, India →Japan, Germany | 1 | CDL (Base Model) | 0.1742 | 0.0225 | 0.0333 | 0.0123 | 0.0120 |
| | | PRUC w/o Causality | **0.2360** | **0.0286** | **0.0721** | **0.0202** | **0.0152** |
| | | PRUC (Full) | 0.1950 | 0.0253 | 0.0677 | 0.0191 | 0.0135 |
| | 2 | CDL (Base Model) | 0.0903 | 0.0102 | 0.0289 | 0.0072 | 0.0054 |
| | | PRUC w/o Causality | 0.1540 | 0.0199 | 0.0570 | 0.0146 | 0.0106 |
| | | PRUC (Full) | **0.1796** | **0.0233** | **0.0579** | **0.0160** | **0.0124** |
| | 3 | CDL (Base Model) | - | - | - | - | - |
| | | PRUC w/o Causality | 0.0692 | 0.0076 | 0.0277 | 0.0059 | 0.0040 |
| | | PRUC (Full) | - | - | - | - | - |
| Germany, Italy, Japan →United States, India | 1 | CDL (Base Model) | 0.0262 | 0.0059 | **0.0079** | 0.0041 | 0.0033 |
| | | PRUC w/o Causality | **0.0263** | **0.0064** | 0.0071 | **0.0049** | **0.0036** |
| | | PRUC (Full) | 0.0262 | **0.0064** | 0.0065 | 0.0042 | **0.0036** |
| | 2 | CDL (Base Model) | **0.0244** | **0.0054** | **0.0088** | **0.0042** | **0.0031** |
| | | PRUC w/o Causality | 0.0205 | 0.0045 | 0.0061 | 0.0032 | 0.0025 |
| | | PRUC (Full) | 0.0208 | 0.0050 | 0.0070 | 0.0036 | 0.0028 |
| | 3 | CDL (Base Model) | **0.0277** | **0.0049** | **0.0066** | **0.0028** | **0.0027** |
| | | PRUC w/o Causality | 0.0101 | 0.0024 | 0.0010 | 0.0011 | 0.0013 |
| | | PRUC (Full) | 0.0137 | 0.0026 | 0.0030 | 0.0016 | 0.0014 |

**Table 4:** Performance of PRUC on different user clusters with DLRM as the base model. "-" means a cluster contains only training-set users, i.e., no test-set users to evaluate. The best results are marked with **bold face**.

| Data | Cluster | Method | Recall@20 | F1@20 | MAP@20 | NDCG@20 | Precision@20 |
|---|---|---|---|---|---|---|---|
| France, Italy, India →Japan, Mexico | 1 | DLRM (Base Model) | 0.0051 | 0.0005 | 0.0004 | 0.0002 | 0.0003 |
| | | PRUC w/o Causality | 0.0137 | 0.0013 | 0.0019 | 0.0006 | 0.0007 |
| | | PRUC (Full) | **0.0345** | **0.004** | 0.0056 | **0.0021** | **0.0021** |
| | 2 | DLRM (Base Model) | 0.0000 | 0.0000 | 0.0000 | 0.0000 | 0.0000 |
| | | PRUC w/o Causality | **0.0208** | **0.0024** | **0.0045** | **0.0014** | **0.0013** |
| | | PRUC (Full) | 0.0000 | 0.0000 | 0.0000 | 0.0000 | 0.0000 |
| | 3 | DLRM (Base Model) | - | - | - | - | - |
| | | PRUC w/o Causality | 0.0334 | 0.0038 | 0.0052 | 0.0021 | 0.0020 |
| | | PRUC (Full) | - | - | - | - | - |
| Mexico, Spain, India →Japan, Germany | 1 | DLRM (Base Model) | 0.0000 | 0.0000 | 0.0000 | 0.0000 | 0.0000 |
| | | PRUC w/o Causality | 0.1621 | 0.0176 | **0.0218** | 0.0085 | 0.0093 |
| | | PRUC (Full) | **0.3074** | **0.0395** | 0.0213 | **0.0152** | **0.0211** |
| | 2 | DLRM (Base Model) | 0.0780 | 0.0096 | 0.0087 | 0.0042 | 0.0051 |
| | | PRUC w/o Causality | 0.1607 | 0.0201 | 0.0363 | 0.0113 | 0.0107 |
| | | PRUC (Full) | **0.1984** | **0.0241** | **0.0555** | **0.0157** | **0.0128** |
| | 3 | DLRM (Base Model) | - | - | - | - | - |
| | | PRUC w/o Causality | 0.1128 | 0.0166 | 0.0245 | 0.0095 | 0.0090 |
| | | PRUC (Full) | - | - | - | - | - |
| Germany, Italy, Japan →United States, India | 1 | DLRM (Base Model) | 0.0023 | 0.0006 | 0.0003 | 0.0003 | 0.0003 |
| | | PRUC w/o Causality | 0.0039 | 0.0010 | **0.0013** | **0.0008** | **0.0006** |
| | | PRUC (Full) | **0.0046** | **0.0011** | 0.0009 | 0.0007 | **0.0006** |
| | 2 | DLRM (Base Model) | 0.0018 | 0.0005 | 0.0003 | 0.0003 | 0.0003 |
| | | PRUC w/o Causality | **0.0053** | **0.0015** | 0.0011 | **0.0009** | **0.0008** |
| | | PRUC (Full) | 0.0045 | 0.0011 | **0.0012** | 0.0007 | 0.0007 |
| | 3 | DLRM (Base Model) | 0.0036 | 0.0008 | 0.0005 | 0.0004 | 0.0004 |
| | | PRUC w/o Causality | 0.0028 | 0.0006 | 0.0003 | 0.0003 | 0.0004 |
| | | PRUC (Full) | **0.0141** | **0.0034** | **0.0075** | **0.0032** | **0.0019** |

ity", demonstrating the causal inference's value in PRUC. Similarly, comparing the performance of the base model and "PRUC w/o Causality", we can see that the functionality of discovering meaningful user clusters does improve the performance.

## 4 Related Work

**Domain-Dependent Recommendation.** Previous work has explored various in-domain recommendation scenarios. Early methods, such as PMF (Mnih & Salakhutdinov, 2007) and BPR (Rendle et al., 2012), applied collaborative filtering techniques to address recommendation challenges. Later, methods such as GRU4Rec (Hidasi et al., 2016), SAS4Rec (Kang & McAuley, 2018) and KGAT (Wang et al., 2019) leveraged advanced deep learning models to enhance the performance of recommender systems. These approaches focus on rating data between items and users but do

**Table 5:** Performance of PRUC on different user clusters with PerK as the base model. "-" means a cluster contains only training-set users, i.e., no test-set users to evaluate. The best results are marked with **bold face**.

| Data | Cluster | Method | Recall@20 | F1@20 | MAP@20 | NDCG@20 | Precision@20 |
|---|---|---|---|---|---|---|---|
| France, Italy, India →Japan, Mexico | 1 | PerK (Base Model) | **0.1752** | **0.0204** | **0.1152** | **0.022** | **0.0108** |
| | | PRUC w/o Causality | 0.0544 | 0.0065 | 0.0084 | 0.0036 | 0.0035 |
| | | PRUC (Full) | 0.1662 | 0.0188 | 0.1087 | 0.0205 | 0.0099 |
| | 2 | PerK (Base Model) | 0.0986 | 0.0115 | 0.0403 | 0.0094 | 0.0061 |
| | | PRUC w/o Causality | **0.1899** | **0.0220** | **0.0841** | **0.01880** | **0.0117** |
| | | PRUC (Full) | 0.1629 | 0.0189 | 0.0548 | 0.0138 | 0.0100 |
| | 3 | PerK (Base Model) | - | - | - | - | - |
| | | PRUC w/o Causality | 0.0000 | 0.0000 | 0.0000 | 0.0000 | 0.0000 |
| | | PRUC (Full) | - | - | - | - | - |
| Mexico, Spain, India →Japan, Germany | 1 | PerK (Base Model) | 0.1434 | 0.0176 | 0.0582 | 0.014 | 0.0094 |
| | | PRUC w/o Causality | 0.2724 | 0.0325 | 0.1064 | 0.0261 | 0.0173 |
| | | PRUC (Full) | **0.2826** | **0.0345** | **0.1152** | **0.0275** | **0.0184** |
| | 2 | PerK (Base Model) | 0.1495 | 0.0184 | 0.0723 | 0.0166 | 0.0098 |
| | | PRUC w/o Causality | **0.2536** | **0.0295** | **0.1082** | **0.0244** | **0.0157** |
| | | PRUC (Full) | 0.1499 | 0.0176 | 0.0651 | 0.0150 | 0.0093 |
| | 3 | PerK (Base Model) | - | - | - | - | - |
| | | PRUC w/o Causality | 0.0530 | 0.0072 | 0.0227 | 0.0063 | 0.0039 |
| | | PRUC (Full) | - | - | - | - | - |
| Germany, Italy, Japan →United States, India | 1 | PerK (Base Model) | 0.0194 | 0.0043 | 0.0057 | 0.003 | 0.0024 |
| | | PRUC w/o Causality | 0.0296 | **0.0070** | **0.0091** | **0.0050** | **0.0039** |
| | | PRUC (Full) | **0.0308** | 0.0068 | 0.0086 | 0.0046 | 0.0038 |
| | 2 | PerK (Base Model) | 0.0126 | 0.0028 | 0.0032 | 0.0018 | 0.0016 |
| | | PRUC w/o Causality | **0.0188** | **0.0039** | 0.0047 | **0.0025** | **0.0022** |
| | | PRUC (Full) | 0.0162 | 0.0037 | **0.0048** | **0.0025** | 0.0021 |
| | 3 | PerK (Base Model) | **0.0261** | **0.0035** | **0.0091** | **0.0025** | **0.0019** |
| | | PRUC w/o Causality | 0.0137 | 0.0031 | 0.0037 | 0.0020 | 0.0018 |
| | | PRUC (Full) | 0.0174 | 0.0027 | 0.0016 | 0.0013 | 0.0015 |

not account for item features. Collaborative deep learning (CDL) models (Wang et al., 2015; 2016; Zhang et al., 2016; Li & She, 2017) incorporate feature data to enable pretrained recommenders, making them more versatile in different contexts, such as cold start scenarios.

Despite significant advances in in-domain recommendations, cross-domain recommendation remains relatively understudied. Existing work has utilized domain adaptation techniques (Xu et al., 2023; Liu et al., 2023; Shi & Wang, 2023; Xu et al., 2022; Wang et al., 2020a; Ganin et al., 2016) to tackle this challenge, often relying on common users or items across source and target domains (Yuan et al., 2020; Wu et al., 2020; Bi et al., 2020; Li et al., 2019; Hansen et al., 2020; Liang et al., 2020; Zhu et al., 2020; Liu et al., 2020). On the other hand, some methods enhance recommendation performance in both source and target domains simultaneously (Li & Tuzhilin, 2020; Hu et al., 2018; Zhao et al., 2019). In contrast to existing approaches, our PRUC first infers the user clusters and confounders, and subsequently makes recommendations based on the identified user clusters, offering better generalization and stronger robustness against domain shifts.

**Causal Inference for Recommendation.** Causal inference (Pearl, 2009) has been widely applied to model cause-and-effect relationships between variables in the machine learning community. Recently, it has been employed to improve the performance of recommender systems (Wang et al., 2020b). PDA (Zhang et al., 2021) uses causal intervention to address popularity bias in recommendations, while DICE (Zheng et al., 2021) learns representations from user interactions based on the structured causal model (SCM). Additionally, some work focuses on debiasing recommendations without a causal inference perspective (Li et al., 2021; Wang et al., 2022; Chen et al., 2023). However, these approaches do not account for user groups in SCM modeling. In contrast, our method divides users into clusters based on a confounder variable and recommends by aggregating users' ratings through do-calculus, offering a more interpretable and sophisticated approach.

## 5 Conclusion

In this paper, we address the problem of cross-domain recommendation by introducing a novel causal Bayesian framework, named Probabilistic Residual User Clustering (PRUC). PRUC generates recommendations by: (1) inferring the user cluster ID, (2) inferring the residual rating based on our causal debiasing framework, and (3) predicting the final rating as a correction to the base model's prediction. PRUC can enhance the performance of any base recommenders in a plug-and-play manner, and automatically discover meaningful user clusters. As a general probabilistic framework compatible with various recommendation systems, PRUC can be extended to additional modalities beyond textual data in future research.

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

# A  Experimental Details

## A.1  Metrics

**mAP.** mAP is defined as:

$$\text{AP}_i = \frac{1}{|J_i|} \sum_{n=1}^{N} \text{rel}_{i,n} \times \text{Precision}_i@n, \tag{30}$$

where $N$ is the total number of recommended items, $\text{Precision}_i@n$ is the precision at rank $n$, and $|J_i|$ is the total number of relevant items for user $i$. The mean Average Precision (mAP) is then calculated by averaging $\text{AP}_i$ over all users:

$$\text{mAP} = \frac{1}{|I|} \sum_{i=1}^{|I|} \text{AP}_i, \tag{31}$$

where $|I|$ is the total number of users.

**NDCG.** NDCG@$N$ is computed as follows:

First, the Discounted Cumulative Gain (DCG@$N$) is calculated:

$$\text{DCG}_i@N = \sum_{n=1}^{N} \frac{2^{\text{rel}_{i,n}} - 1}{\log_2(n+1)}, \tag{32}$$

where $\text{rel}_{i,n}$ denotes the relevance of the item at position $n$ for user $i$. Next, the Ideal Discounted Cumulative Gain (IDCG@$N$), representing the maximum possible DCG (i.e., all relevant items ranked at the top), is calculated as:

$$\text{IDCG}_i@N = \sum_{n=1}^{\min(N,|J_i|)} \frac{2^1 - 1}{\log_2(n+1)} = \sum_{n=1}^{\min(N,|J_i|)} \frac{1}{\log_2(n+1)}, \tag{33}$$

where $|J_i|$ denotes the total number of relevant items for user $i$.

Finally, the Normalized Discounted Cumulative Gain is obtained by normalizing DCG@$N$ by IDCG@$N$:

$$\text{NDCG}_i@N = \frac{\text{DCG}_i@N}{\text{IDCG}_i@N}. \tag{34}$$

Here the logarithmic term $\log_2(n+1)$ discounts the relevance based on the item's position in the ranked list, serving as the normalization factor.

## A.2 Training Configurations

We set the hidden dimension $h = 50$ for all latent vectors, as well as for the encoder and decoder networks. During training, we use AdamW (Kingma & Ba, 2015) as our optimizer, with a learning rate of $10^{-3}$ and a batch size of $256$. The base models were trained for 100 epochs, while PRUC was trained for 150 epochs. All experiments were conducted on an NVIDIA RTX A5000 GPU.

## A.3 Explanation of the Cluster

Figure 5 illustrates the relationship between user clusters and items, as inferred by the CDL-based PRUC model. For each user, we selected the item with the highest rating, recorded its rating, and visualized the results. Different clusters are represented using distinct colors, effectively showcasing the distribution and preferences of users within each cluster. The figure shows that different clusters

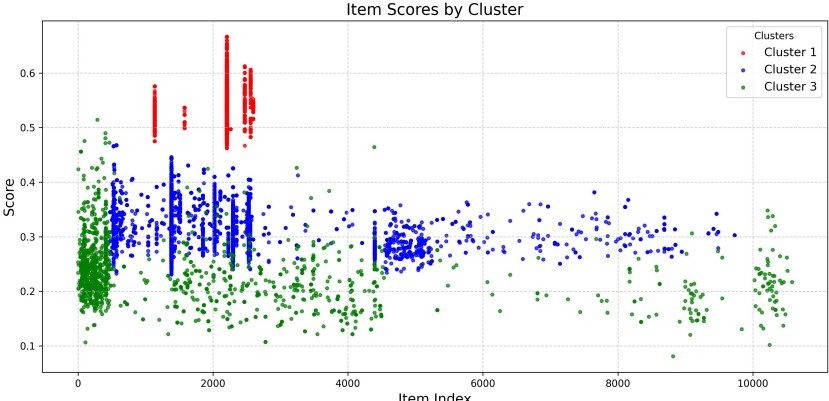

**Figure 5:** the explanation of cluster

represent distinct preferences of users for items. For example, Cluster 1 (Red) exhibits more focused preferences for certain items with distinct item indices. This finding effectively explains the effect of user clustering in enhancing the performance of PRUC's recommender.

