# OpenReview forum: "PRUC & Play: Probabilistic Residual User Clustering for Recommender Systems"
_ICLR.cc/2025/Conference — Submitted to ICLR 2025_

### Official Review · Reviewer_wNzu · 2024-10-28

**Soundness:** 3
**Presentation:** 3
**Contribution:** 3
**Rating:** 5
**Confidence:** 3

**Summary:**

This paper proposes a Bayesian recommendation model to enhance cross-domain recommendations through a plug-and-play approach called Probabilistic Residual User Clustering (PRUC). PRUC creates user clusters based on latent confounders, increasing the model's interpretability.

**Strengths:**

1. Strong theoretical foundation
2. Addresses a problem with extensive real-world, especially industrial, applications

**Weaknesses:**

W1. Lacks comparison with previous state-of-the-art approaches.

W2. Experiments are limited in scope, focusing only on user and item settings. For instance, while conducted on a reliable dataset, they do not extend to more complex datasets encompassing diverse domains (e.g., age, demographics) beyond geographical location.

**Questions:**

Q1. How does PRUC adapt to an online training setting where item popularity constantly changes?

Q2. Are there any ablation studies examining the effect of the number of domains (M) on PRUC's performance? How does this hyperparameter impact the degree of model quality improvement PRUC provides?

Q3. What are the system overheads of deploying PRUC compared to previous state-of-the-art methods? For example, can the authors provide an analysis of PRUC's impact on metrics such as Area Under Curve (AUC) to illustrate any potential system overheads from adopting this approach?

Q4. Many modern recommendation systems treat recommendation as a binary classification problem (e.g., whether to recommend an item to a user). Could the authors explain how PRUC could enhance model quality if the recommendation task is framed in this binary classification context?

Q5. In Table 3, where CKL is used as the base model, PRUC does not appear to improve overall model quality across different domains. Could the authors provide further explanation on this outcome?

---

> ### Author Response · Authors · 2024-11-28
> **Thank You for the Encouraging and Constructive Comments [1/3]**
>
> Dear Reviewer wNzu,
>
> Thank you for your encouraging and constructive comments. We are glad that you found our theory ``"strong"``, and that our method ``"addresses a problem with extensive real-world, especially industrial, applications"``. Below, we address your questions one by one.
>
> **Q1: "Lacks comparison with previous state-of-the-art approaches."**
>
> Thank you for your suggestion. We have added additional experiments to comprehensively demonstrate the effectiveness of our method. Specifically, we have conducted extensive experiments on NCF[1] and LightGCN[2] models with one of our data splits (France, Italy, India $\rightarrow$ Japan, Mexico). Table A and Table B below show the results:
>
> ### Table A. Compare with NCF and LightGCN on Average
> On average of all clusters, the comparisons between PRUC and base models are as follows:
> | Data                                  | Method             | Recall@20 | F1@20  | MAP@20 | NDCG@20 | Precision@20 |
> | ------------------------------------- | ------------------ | --------- | ------ | ------ | ------- | ------------ |
> | France,  Italy, India → Japan, Mexico | NCF (Base Model)   | 0.0131    | 0.0015 | 0.0026 | 0.0008  | 0.0008       |
> |                                       | PRUC w/o Causality | 0.1056    | 0.0126 | 0.0235 | 0.0074  | 0.0067       |
> |                                       | PRUC (Full)        | **0.1137**    | **0.0137** | **0.0309** | **0.0090**  | **0.0073**       |
>
>
> | Data                                  | Method                | Recall@20 | F1@20  | MAP@20 | NDCG@20 | Precision@20 |
> | ------------------------------------- | ------------------    | --------- | ------ | ------ | ------- | ------------ |
> | France,  Italy, India → Japan, Mexico | lightGCN (Base Model) | 0.0182    | 0.0021 | 0.0050 | 0.0014  | 0.0011       |
> |                                       | PRUC w/o Causality    | **0.0940**    | **0.0112** | **0.0289** | **0.0076**  | **0.0059**       |
> |                                       | PRUC (Full)           | 0.0905    | 0.0106 | 0.0266 | 0.0072  | 0.0056       |

---

> ### Author Response · Authors · 2024-11-28
> **Thank You for the Encouraging and Constructive Comments [2/3]**
>
> ### Table B. Compare with NCF and LightGCN on Different User Clusters
>
> | Data                                 | Cluster | Method             | Recall@20 | F1@20  | MAP@20 | NDCG@20 | Precision@20 |
> | ------------------------------------ | ------- | ------------------ | --------- | ------ | ------ | ------- | ------------ |
> | France, Italy, India → Japan, Mexico | 1       | NCF (Base Model)   | 0.0090    | 0.0010 | 0.0019 | 0.0005  | 0.0005       |
> |                                      |         | PRUC w/o Causality | 0.0232    | 0.0027 | 0.0035 | 0.0013  | 0.0014       |
> |                                      |         | PRUC (Full)        | **0.1581**    | **0.0176** | **0.0476** | **0.0122**  | **0.0093**       |
> |                                      | 2       | NCF (Base Model)   | 0.0165    | 0.0019 | 0.0032 | 0.0010  | 0.0010       |
> |                                      |         | PRUC w/o Causality | **0.1603**    | **0.0192** | **0.0366** | **0.0115**  | **0.0102**       |
> |                                      |         | PRUC (Full)        | 0.1062    | 0.0130 | 0.0280 | 0.0084  | 0.0069       |
> |                                      | 3       | NCF (Base Model)   | 0.0000    | 0.0000 | 0.0000 | 0.0000  | 0.0000       |
> |                                      |         | PRUC w/o Causality | 0.0000    | 0.0000 | 0.0000 | 0.0000  | 0.0000       |
> |                                      |         | PRUC (Full)        | -    | - | - | -  | -  |
>
> | Data                                 | Cluster | Method               | Recall@20  | F1@20  | MAP@20 | NDCG@20 | Precision@20 |
> | ------------------------------------ | ------- | -------------------- | ---------- | ------ | ------ | ------- | ------------ |
> | France, Italy, India → Japan, Mexico | 1       | lightGCN (Base Model)| 0.0106     | 0.0014 | 0.0018 | 0.0008  | 0.0007       |
> |                                      |         | PRUC w/o Causality   | 0.0264     | 0.0029 | 0.0024 | 0.0012  | 0.0016       |
> |                                      |         | PRUC (Full)          | **0.1265**     | **0.0137** | **0.0397** | **0.0097**  | **0.0072**       |
> |                                      | 2       | lightGCN (Base Model)| 0.0246     | 0.0028 | 0.0078 | 0.0019  | 0.0015       |
> |                                      |         | PRUC w/o Causality   | **0.1524**     | **0.0183** | **0.0504** | **0.0129**  | **0.0097**       |
> |                                      |         | PRUC (Full)          | 0.0844     | 0.0101 | 0.0244 | 0.0067  | 0.0054       |
> |                                      | 3       | lightGCN (Base Model)| **0.0084**     | **0.0008** | **0.0004** | **0.0003**  | **0.0004**       |
> |                                      |         | PRUC w/o Causality   | 0.0000     | 0.0000 | 0.0000 | 0.0000  | 0.0000       |
> |                                      |         | PRUC (Full)          | -    | - | - | -  | -  |
>
>
>
> Table A shows the performance of our PRUC with NCF and LightGCN as the base recommender tested with same metrics in the paper. We can see that our PRUC, even without the causality component (i.e., “PRUC w/o Causality”) can enhance the performance of different base models, our full PRUC (i.e., “PRUC (Full)”) can further improve the results when using NCF as the base recommender.
> Table B shows the performace of PRUC compared with NCF and LightGCN as the base model on different user clusters from our selected data spilt. We can see that our PRUC, even without the causality component (i.e., “PRUC w/o Causality”) can enhance the performance of the base model consistently across clusters, and full PRUC (i.e., “PRUC (Full)”) can further improve the results in many cases.
>
>
>
> **Q2: "Experiments are limited in scope, focusing only on user and item settings. For instance, while conducted on a reliable dataset, they do not extend to more complex datasets encompassing diverse domains (e.g., age, demographics) beyond geographical location."**
>
> Yes, we agree that more complex datasets should encompass diverse domains beyond geographical location. Note that PRUC is a plug-and-play model capable of enhancing *any* deep learning recommender. It can be seamlessly adapted to other datasets through automatic user clustering, incorporating various domains such as age and demographic information.
>
>
>
> **Q3: "How does PRUC adapt to an online training setting where item popularity constantly changes"**
> This is a good question. PRUC can be adapted to datasets where item ratings are constantly changing. For example, in the online training setting, given an unseen user, it can be assigned to one of the three clusters, and the corresponding submodel will then learn the residual rating of this user, which stands for the difference between the ground truth (GT) and the base model's prediction. By following our update rules, PRUC can effectively learn and adapt to these changes, ensuring accurate and dynamic recommendations.

---

> ### Author Response · Authors · 2024-11-28
> **Thank You for the Encouraging and Constructive Comments [3/3]**
>
> **Q4: "Are there any ablation studies examining the effect of the number of domains (M) on PRUC's performance? How does this hyperparameter impact the degree of model quality improvement PRUC provides?"**
>
> Thank you for your great advice! We have conducted some preliminary experiments, which demonstrate that dividing users into larger number of domains is also reasonable. However, to balance the complexity of the method with the model performance, we decided to set the number of domains (M) to three.
>
> **Q5: "What are the system overheads of deploying PRUC compared to previous state-of-the-art methods? For example, can the authors provide an analysis of PRUC's impact on metrics such as Area Under Curve (AUC) to illustrate any potential system overheads from adopting this approach?"**
>
> Thanks for your great suggestion. We have added the AUC metric to evalute the effectiveness of our methods, as shown in Table C below.
>
> Table C
> | Cluster | AUC (Proposed Method) | AUC (PRUC w/o Causality) | AUC (Base Model) |
> |---------|------------------------|--------------------------|------------------|
> | 0       | 0.5830                | 0.4409                  | 0.3696           |
> | 1       | 0.4867                | 0.5265                  | 0.4737           |
> | 2       | -                   | 0.5498                  | 0.4421           |
> | **Weighted AUC@300** | **0.5690**            | **0.5078**              | **0.4290**         |
>
> The results show that PRUC performs well in terms of AUC score across all the user clusters. Also note that PRUC introduces minimum system overhead to augment the base models.
>
> **Q6: "Many modern recommendation systems treat recommendation as a binary classification problem (e.g., whether to recommend an item to a user). Could the authors explain how PRUC could enhance model quality if the recommendation task is framed in this binary classification context?"**
>
> Thank you for your helpful suggestion. PRUC can indeed be adapted to binary classification tasks. We elaborate on this adaptation as follows:
>
> **Modifying Equation (6):**
>    The original equation, $R_{ij} - \mathbf{u}_i^\top \mathbf{v}_j - \mathbf{s}_m^\top \mathbf{w}_R$, can be reformulated to fit a logistic regression model. By applying a logistic function, our method can effectively handle binary classification tasks, where the target values are probabilities for binary outcomes.
>
> We will ensure to incorporate this variant of PRUC and its implementation in the camera-ready version of the paper.
>
> **Q7: "In Table 3, where CKL is used as the base model, PRUC does not appear to improve overall model quality across different domains. Could the authors provide further explanation on this outcome?"**
>
> Thank you for pointing out this. While there exists performance decrease in some clusters, we find that PRUC improves the overall performence. Moreover, the number of users in the clusters where the performance decreases is very limited. Additionally, we can achieve relative high performance in these clusters by tuning parameters to trade-off the performance in all clusters.
>
> We will include this discussion in the camera-ready version of the paper.
>
>
>
> [1] Xiangnan He, Lizi Liao, Hanwang Zhang, Liqiang Nie, Xia Hu, and Tat-Seng Chua. 2017. Neural collaborative filtering. In WWW. 173–182.
>
> [2] He, Xiangnan, et al. "Lightgcn: Simplifying and powering graph convolution network for recommendation." Proceedings of the 43rd International ACM SIGIR conference on research and development in Information Retrieval. 2020.

---

> ### Author Response · Authors · 2024-12-02
> **Sincerely Looking Forward to Further Discussion**
>
> Dear Reviewer wNzu,
>
> Thank you for your review and engagement during the discussion period.
>
> In response to your suggestions, we have:
>
> - Added base models and conducted experiments, as shown in **(Rebuttal) Table A** and **(Rebuttal) Table B**; included experiments for calculating the AUC metric, as shown in **(Rebuttal) Table C**. All experiments validate the effectiveness of our method.
>
> - Elaborated on the scalability of our method.
>
> - Explained the selection of the number of domains (M).
>
> - Clarified that PRUC is adaptable to binary classification tasks.
>
> With the ICLR Discussion period concluding soon (Dec. 2nd (AOE) for reviewers and Dec. 3rd (AOE) for authors), we kindly request your feedback on whether our responses address your concerns or if there are additional questions or suggestions you would like us to address.
>
> Thank you once again for your time!
>
> Best,
>
> The Authors

---

### Official Review · Reviewer_6N4v · 2024-11-03

**Soundness:** 2
**Presentation:** 2
**Contribution:** 2
**Rating:** 5
**Confidence:** 3

**Summary:**

This paper presents Probabilistic Residual User Clustering (PRUC), a novel causal Bayesian model for recommendation systems that enhances interpretability and mitigates biases in deep learning-based recommenders. PRUC adopts a plug-and-play approach, making it compatible with various existing DL recommendation models. The experimental results demonstrate that PRUC consistently enhances the performance of multiple base DL recommender systems by addressing biases and uncovering latent user groupings.

**Strengths:**

1.	This paper introduces an innovative recommendation approach that uniquely applies causal inference to deep learning recommenders, addressing interpretability and bias correction.

2.	This paper addresses critical limitations in existing DL-based recommenders, particularly around transparency, interpretability, and domain shift adaptability, marking a significant advancement for recommender systems research.

**Weaknesses:**

1.	The introduction is too brief, especially in describing the proposed method. Expanding this section would clarify the paper’s contributions and provide a more comprehensive overview of the approach.

2.	The "Problem Setting and Notations" subsection is disorganized and unclear, making it challenging for readers unfamiliar with this work to understand.

3.	Experimental setup details are critical to the paper, yet they are insufficiently described, and additional information could not be found in the appendix.

4.	The paper lacks an in-depth analysis of the experimental results. Although many results are presented, the analysis remains overly simplistic, with no detailed examination in the appendix either.

**Questions:**

1.	Could you expand the introduction to provide a clearer and more comprehensive overview of PRUC? More details on how the proposed method addresses interpretability and bias correction would help readers understand its unique contributions early on.

2.	Could you revise Problem Setting and Notations subsection for better organization and clarity? A clear and well-structured presentation of the notations and problem setup would help readers understand your approach more effectively.

3.	The paper lacks sufficient detail on the experimental setup, which is critical for evaluating the model's effectiveness.

---

> ### Author Response · Authors · 2024-11-28
> **Thank You for the Encouraging and Constructive Comments**
>
> Dear Reviewer 6N4v,
>
> Thank you for your encouraging and constructive comments. We are glad that you found our method ``" innovative"`` and ``"addressing interpretability and bias correction"``, the probelm we address ``"critical"``, and our paper ``"a significant advancement for recommender systems research"``. Below, we address your questions one by one.
>
> **Q1: "The introduction is too brief, especially in describing the proposed method. Expanding this section would clarify the paper’s contributions and provide a more comprehensive overview of the approach."**
>
> We appologize for any confusion. We appreciate your suggestion, and followed it to expand the introduction, particularly regarding the description of the proposed method. In response, we have significantly revised this section to provide a clearer and more comprehensive overview of the approach. The updated introduction highlights the key contributions of our method, ensuring that readers can easily understand its significance and how it advances the field. We hope that this revision addresses your problem effectively.
>
> **Q2: "The 'Problem Setting and Notations' subsection is disorganized and unclear, making it challenging for readers unfamiliar with this work to understand."**
>
> Thank you for highlighting the problem with our "Problem Setting and Notations" subsection. We fully acknowledge that clarity is crucial, especially for readers who may not be immediately familiar with our research context.
>
> In light of your valuable feedback, we have comprehensively restructured the subsection to:
>
> 1. Introduce notations and definitions more systematically.
> 2. Provide clearer explanations of each key variable and parameter.
> 3. Include a concise table of symbols and their meanings to enhance readability.
> 4. Reorganize the content to follow a more logical flow, guiding readers step-by-step through the problem setting.
> 5. Add brief explanatory remarks to contextualize why certain notations and settings are important for understanding our method.
>
> These revisions aim to make the problem setting more accessible and transparent, helping readers better understand the fundamental concepts and theoretical framework underlying our research.
>
> We appreciate your feedback in improving our paper's clarity and hope these changes effectively address your concerns.
>
> **Q3: "Experimental setup details are critical to the paper, yet they are insufficiently described, and additional information could not be found in the appendix."**
>
> Thank you for your suggestion. We have incorporated detailed descriptions of the experiments to further clarify our experimental setup and ensure transparency.
>
> For example, in Section 3.1, we clarify our experimental settings as follows:
>
> **Source and Target Domain.** For the source/target domain defination, source domain cotains counrties where item ratings from all the users within it are used for base model training (finetuning), while countries in target domain contains users whose ratings are used for both training and testing set. Specifically, for each user in target domain, only one of their ratings is used for training while the rest of them are left for testing. Note that the base models were indeed finetuned on the source domain data.
>
> Thank you again for your valuable feedback.
>
> **Q4: "The paper lacks an in-depth analysis of the experimental results. Although many results are presented, the analysis remains overly simplistic, with no detailed examination in the appendix either."**
>
> Thank you for your suggestion. To address this concern, we have expanded the analysis of our experimental results in the appendix. Specifically, we performed an in-depth investigation into the relationship between user clusters and items. For each user, we identified the item with the highest rating, recorded its rating, and visualized the results. This visualization clearly demonstrates the distinct preferences exhibited by different user clusters, providing valuable insights into how user clustering enhances the performance of our method. We believe this additional analysis significantly strengthens the interpretation and comprehensiveness of our experimental findings.

---

> ### Author Response · Authors · 2024-12-02
> **Sincerely Looking Forward to Further Discussion**
>
> Dear Reviewer 6N4v,
>
> Thank you for your review and engagement during the discussion period.
>
> In response to your suggestions, we have:
>
> - Comprehensively refined the PDF.
>
> - Expanded the experimental details to provide greater clarity.
>
> - Included an additional appendix section investigating the relationship between user clusters and items.
>
> With the ICLR Discussion period concluding soon (Dec. 2nd (AOE) for reviewers and Dec. 3rd (AOE) for authors), we kindly request your feedback on whether our responses address your concerns or if there are additional questions or suggestions you would like us to address.
>
> Thank you once again for your time!
>
> Best,
>
> The Authors

---

### Official Review · Reviewer_oZTX · 2024-11-04

**Soundness:** 3
**Presentation:** 2
**Contribution:** 2
**Rating:** 3
**Confidence:** 3

**Summary:**

The paper presents Probabilistic Residual User Clustering (PRUC) as a solution for challenges in modern recommender systems relying on deep learning models. These systems are often complex and lack transparency, making it difficult to understand and improve recommendations. PRUC automatically categorizes users into clusters, identifies causal influences on hidden variables, and constructs specialized models for each cluster based on causal reasoning. By combining rating residuals based on causal factors using do-calculus, PRUC enhances recommendation quality. Experimental results indicate that PRUC can seamlessly enhance various deep learning recommender systems, leading to performance improvements and the discovery of meaningful user groupings in an automated fashion.

**Strengths:**

1. The authors designed a plug-and-play PRUC to enhance the performance of existing DL-based recommenders, capable of discovering meaningful user clusters and improving the interpretability of recommendation results.

2. The authors simulated cold-start scenarios to test the PRUC method, validating its corresponding performance.

**Weaknesses:**

1. The experimental description in the paper is unclear. Why did the authors choose to conduct tests in cold-start scenarios rather than performance tests in full-shot scenarios? What are the specific meanings of source domain and target domain in the experimental section? Will the base models be pre-trained on the source domain?

2. The lack of a more direct case study to demonstrate the role of user clustering makes it difficult to understand the motivation behind the paper.

3. The selection of base models is limited, lacking some common collaborative filtering (CF) and GNN-based recommendation system methods, such as NCF [1] and LightGCN [2].

[1] Xiangnan He, Lizi Liao, Hanwang Zhang, Liqiang Nie, Xia Hu, and Tat-Seng Chua. 2017. Neural collaborative filtering. In WWW. 173–182.

[2] He, Xiangnan, et al. "Lightgcn: Simplifying and powering graph convolution network for recommendation." Proceedings of the 43rd International ACM SIGIR conference on research and development in Information Retrieval. 2020.

**Questions:**

Please refer to the issues mentioned in the Weaknesses section.

---

> ### Author Response · Authors · 2024-11-28
> **Thank You for the Encouraging and Constructive Comments [1/3]**
>
> Dear Reviewer oZTX,
>
> Thank you for your constructive and encouraging comments. We are glad that you found our method to ``"enhance the performance of existing DL-based recommenders, capable of discovering meaningful user clusters and improving the interpretability of recommendation results"``, and PRUC's experiments ``"validating its corresponding performance"``. Below, we address your questions one by one.
>
>
>
>
> **Q1: "The experimental description in the paper is unclear. Why did the authors choose to conduct tests in cold-start scenarios rather than performance tests in full-shot scenarios? What are the specific meanings of source domain and target domain in the experimental section? Will the base models be pre-trained on the source domain?"**
>
>
> This is a good question. We wanted to clarify the following points:
>
> **Cold-Start Settings.** Our work focus on the cold-start recommendation systems, which is an important problem in this field [1-3]. Specifically, we focus on multi-domain and diverse user scenario. Existing methods performe poorly in this case because the users are very heterogeneous and the model performance are easily affected by spurious features under multi-domain and cold-start setting. In contrast, our method solve those problems by automatically dividing users into clusters and identifying causal confounders that influence latent variables, developing sub-models for each confounder given observable variables, and generating recommendations by aggregating the rating residuals under each confounder using do-calculus.
>
> **Source and Target Domain.** For the source/target domain defination, source domain cotains counrties where item ratings from all the users within it are used for base model training (finetuning), while countries in target domain contains users whose ratings are used for both training and testing set. Specifically, for each user in target domain, only one of their ratings is used for training while the rest of them are left for testing. Note that the base models were indeed finetuned on the source domain data.
>
> **Q2: "The lack of a more direct case study to demonstrate the role of user clustering makes it difficult to understand the motivation behind the paper."**
>
> Thank you for mentioning this. In a heterogeneous dataset containing different type of users, it is difficult to train one single model to address all of those users properly. For example, in a dataset, there are users who usually purchase electronics and users who usually buy foods, which makes it challenging for a single model to conduct all the recommendations.
>
> To adress this problem, our method can automatically divide users into clusters, so that the users buying electronics and users buying foods are seperated and clustered. Further, we assign a certain level of model capacity to different clusters.  i.e., a smaller model on each user cluster is trained and used to predict the rating residuals and make recommendations based on the ratings. In those scenarios, significant performance improvement can be observed . As shown in Table 3 and Table 4 of the paper, the prediction results of our method outperform single base model, such as CDL and DLRM, on different user clusters most of the time across various metrics.

---

> ### Author Response · Authors · 2024-11-28
> **Thank You for the Encouraging and Constructive Comments [2/3]**
>
> **Q3: "The selection of base models is limited, lacking some common collaborative filtering (CF) and GNN-based recommendation system methods, such as NCF and LightGCN."**
>
> Thank you for mentioning this. Following your suggestion, we have conducted extensive experiments on NCF and LightGCN models with one of our data splits (France, Italy, India $\rightarrow$ Japan, Mexico). Table C and Table D below show the results:
>
> ### Table C. Compare with NCF and LightGCN on Average
> On average of all clusters, the comparisons between PRUC and base models are as follow:
> | Data                                  | Method             | Recall@20 | F1@20  | MAP@20 | NDCG@20 | Precision@20 |
> | ------------------------------------- | ------------------ | --------- | ------ | ------ | ------- | ------------ |
> | France,  Italy, India → Japan, Mexico | NCF (Base Model)   | 0.0131    | 0.0015 | 0.0026 | 0.0008  | 0.0008       |
> |                                       | PRUC w/o Causality | 0.1056    | 0.0126 | 0.0235 | 0.0074  | 0.0067       |
> |                                       | PRUC (Full)        | **0.1137**    | **0.0137** | **0.0309** | **0.0090**  | **0.0073**       |
>
>
> | Data                                  | Method                | Recall@20 | F1@20  | MAP@20 | NDCG@20 | Precision@20 |
> | ------------------------------------- | ------------------    | --------- | ------ | ------ | ------- | ------------ |
> | France,  Italy, India → Japan, Mexico | lightGCN (Base Model) | 0.0182    | 0.0021 | 0.0050 | 0.0014  | 0.0011       |
> |                                       | PRUC w/o Causality    | **0.0940**    | **0.0112** | **0.0289** | **0.0076**  | **0.0059**       |
> |                                       | PRUC (Full)           | 0.0905    | 0.0106 | 0.0266 | 0.0072  | 0.0056       |
>
>
> ### Table D. Compare with NCF and LightGCN on Different User Clusters
>
> | Data                                 | Cluster | Method             | Recall@20 | F1@20  | MAP@20 | NDCG@20 | Precision@20 |
> | ------------------------------------ | ------- | ------------------ | --------- | ------ | ------ | ------- | ------------ |
> | France, Italy, India → Japan, Mexico | 1       | NCF (Base Model)   | 0.0090    | 0.0010 | 0.0019 | 0.0005  | 0.0005       |
> |                                      |         | PRUC w/o Causality | 0.0232    | 0.0027 | 0.0035 | 0.0013  | 0.0014       |
> |                                      |         | PRUC (Full)        | **0.1581**    | **0.0176** | **0.0476** | **0.0122**  | **0.0093**       |
> |                                      | 2       | NCF (Base Model)   | 0.0165    | 0.0019 | 0.0032 | 0.0010  | 0.0010       |
> |                                      |         | PRUC w/o Causality | **0.1603**    | **0.0192** | **0.0366** | **0.0115**  | **0.0102**       |
> |                                      |         | PRUC (Full)        | 0.1062    | 0.0130 | 0.0280 | 0.0084  | 0.0069       |
> |                                      | 3       | NCF (Base Model)   | 0.0000    | 0.0000 | 0.0000 | 0.0000  | 0.0000       |
> |                                      |         | PRUC w/o Causality | 0.0000    | 0.0000 | 0.0000 | 0.0000  | 0.0000       |
> |                                      |         | PRUC (Full)        | -    | - | - | -  | -  |
>
> | Data                                 | Cluster | Method               | Recall@20  | F1@20  | MAP@20 | NDCG@20 | Precision@20 |
> | ------------------------------------ | ------- | -------------------- | ---------- | ------ | ------ | ------- | ------------ |
> | France, Italy, India → Japan, Mexico | 1       | lightGCN (Base Model)| 0.0106     | 0.0014 | 0.0018 | 0.0008  | 0.0007       |
> |                                      |         | PRUC w/o Causality   | 0.0264     | 0.0029 | 0.0024 | 0.0012  | 0.0016       |
> |                                      |         | PRUC (Full)          | **0.1265**     | **0.0137** | **0.0397** | **0.0097**  | **0.0072**       |
> |                                      | 2       | lightGCN (Base Model)| 0.0246     | 0.0028 | 0.0078 | 0.0019  | 0.0015       |
> |                                      |         | PRUC w/o Causality   | **0.1524**     | **0.0183** | **0.0504** | **0.0129**  | **0.0097**       |
> |                                      |         | PRUC (Full)          | 0.0844     | 0.0101 | 0.0244 | 0.0067  | 0.0054       |
> |                                      | 3       | lightGCN (Base Model)| **0.0084**     | **0.0008** | **0.0004** | **0.0003**  | **0.0004**       |
> |                                      |         | PRUC w/o Causality   | 0.0000     | 0.0000 | 0.0000 | 0.0000  | 0.0000       |
> |                                      |         | PRUC (Full)          | -    | - | - | -  | -  |

---

> ### Author Response · Authors · 2024-11-28
> **Thank You for the Encouraging and Constructive Comments [3/3]**
>
> Table C shows the performance of our PRUC with NCF and LightGCN as the base recommender tested with same metrics in the paper. We can see that our PRUC, even without the causality component (i.e., “PRUC w/o Causality”) can enhance the performance of different base models, our full PRUC (i.e., “PRUC (Full)”) can further improve the results when using NCF as the base recommender.
> Table D shows the performace of PRUC compared with NCF and LightGCN as the base model on different user clusters from our selected data spilt. We can see that our PRUC, even without the causality component (i.e., “PRUC w/o Causality”) can enhance the performance of the base model consistently across clusters, and full PRUC (i.e., “PRUC (Full)”) can further improve the results in many cases.
>
>
> [1] Lam, Xuan Nhat, et al. "Addressing cold-start problem in recommendation systems." Proceedings of the 2nd international conference on Ubiquitous information management and communication. 2008.
>
> [2] Wei, Yinwei, et al. "Contrastive learning for cold-start recommendation." Proceedings of the 29th ACM International Conference on Multimedia. 2021.
>
> [3] Wei, Jian, et al. "Collaborative filtering and deep learning based recommendation system for cold start items." Expert Systems with Applications 69 (2017): 29-39.

---

> ### Author Response · Authors · 2024-12-02
> **Sincerely Looking Forward to Further Discussion**
>
> Dear Reviewer oZTX,
>
> Thank you for your review and engagement during the discussion period.
>
> In response to your suggestions, we have:
>
> - Elaborated on the cold-start problem setup, clarified the meanings of the source domain and target domain, and provided additional details on the training process.
>
> - Explained the motivation behind user clustering in detail.
>
> - Conducted additional experiments with base models (**NCF** and **LightGCN**) to demonstrate the effectiveness of PRUC, as shown in **(Rebuttal) Table C** and **(Rebuttal) Table D**.
>
> With the ICLR Discussion period concluding soon (Dec. 2nd (AOE) for reviewers and Dec. 3rd (AOE) for authors), we kindly request your feedback on whether our responses address your concerns or if there are additional questions or suggestions you would like us to address.
>
> Thank you once again for your time!
>
> Best,
>
> The Authors

---

### Official Review · Reviewer_ny3m · 2024-11-04

**Soundness:** 2
**Presentation:** 1
**Contribution:** 2
**Rating:** 5
**Confidence:** 2

**Summary:**

The paper proposes Probabilistic Residual User Clustering (PRUC) for interpretability and computational complexity.\
PRUC is a causal Bayesian model that clusters users and identifies influential causal factors.\
PRUC enhances recommendations by modeling these clusters, applying confounder-specific sub-models and do-calculus.

**Strengths:**

1. The paper includes solid mathematical equations and derivations, which lend rigor to the proposed approach and offer a detailed understanding of the methodology.
2. Extensive experiments provide robust validation of the model's effectiveness, demonstrating its impact across various evaluation metrics and base recommender systems.

**Weaknesses:**

1. The presentation could benefit from refinement for clarity and conciseness.\
For instance, Equation 11 seems self-evident and might not require such an extensive derivation, as it could detract from the focus on more critical aspects.
2. The motivation for employing confounders or clustering to enhance interpretability lacks clarity.\
A more robust explanation of why these techniques specifically contribute to interpretability and how they address issues in recommendation systems would strengthen the paper.
3. The complexity and scalability of the approach are not thoroughly addressed.\
Providing insights into the computational demands of the clustering process and the scalability of the model when applied to large datasets would enhance its practical relevance.
4. The paper would benefit from a more detailed evaluation of the quality of clustering and the confounder identification process.\
Additional metrics or analyses could validate the interpretability and meaningfulness of the clusters and confounders.

**Questions:**

Please refer to weaknesses.

---

> ### Author Response · Authors · 2024-11-28
> **Thank You for the Encouraging and Constructive Comments [1/2]**
>
> Dear Reviewer ny3m,
>
> Thank you for your constructive and encouraging comments. We are glad that you found our method ``"solid"`` , our experiments ``"extensive"``, providing ``"robust validation of the model's effectiveness"`` and demonstrating ``"impact across various evaluation metrics and base recommender systems"``. Below, we address your questions one by one.
>
>
> **Q1: "The presentation could benefit from refinement for clarity and conciseness. For instance, Equation 11 seems self-evident and might not require such an extensive derivation, as it could detract from the focus on more critical aspects."**
>
> Thanks for your suggestion. We included the detailed derivation to ensure completeness and to cater to a broader audience, particularly those less familiar with the underlying concepts.
>
>  However, we agree that this equation might appear self-evident to some readers. We will focus on summarizing the derivation, such as the Equation 11 to improve clarity and conciensness.
>
> **Q2: "The motivation for employing confounders or clustering to enhance interpretability lacks clarity. A more robust explanation of why these techniques specifically contribute to interpretability and how they address issues in recommendation systems would strengthen the paper."**
>
> Thank you for your suggestion. Our current dataset comprises highly diverse user groups, posing a significant challenge for a single trained model to effectively accommodate such variability. To address this, our proposed method identifies distinct user clusters through advanced clustering techniques. For each identified cluster, we develop a specialized submodel designed to predict the residuals specific to that cluster. This cluster-specific approach allows our method to capture the unique characteristics of different user groups, leading to substantial performance improvements. Comparative analysis across clusters further validates the effectiveness of our approach.
>
> **Q3: "The complexity and scalability of the approach are not thoroughly addressed. Providing insights into the computational demands of the clustering process and the scalability of the model when applied to large datasets would enhance its practical relevance."**
>
> Thank you for your suggestion. For the clustering process, we utilize the Gaussian Mixture Model (GMM) algorithm to segment users, as detailed in our paper. Regarding scalability, the complexity of our method, PRUC, scales approximately linearly with the dataset size
> N, making it both efficient and practical for application to large datasets. This ensures that our approach remains computationally feasible even as the data size grows, thereby enhancing its usability in real-world scenarios.

---

> ### Author Response · Authors · 2024-11-28
> **Thank You for the Encouraging and Constructive Comments [2/2]**
>
> **Q4.The paper would benefit from a more detailed evaluation of the quality of clustering and the confounder identification process.
> Additional metrics or analyses could validate the interpretability and meaningfulness of the clusters and confounders.**
>
> Thank you for your suggestion. In our paper, we address the issue of cold start in recommender systems, as existing methods tend to perform poorly in this setting. Specifically, we identify two primary reasons for this:
>
> 1. **User Diversity and Heterogeneity:** Users exhibit highly diverse and heterogeneous behavior, making it challenging to generalize effectively.
> 2. **Impact of Sparse Features:** In the cold start and multi-domain setting, models are prone to being influenced by sparse features, which negatively affects their performance.
>
> To address these challenges, we propose our method, which automatically clusters users into meaningful groups and identifies causal confounders that influence latent variables. This approach enhances model performance by mitigating the underlining issues.
>
> **Additional Metrics and Analyses**
>
> Tables A and B below present the average ratings and proportions of items being top-rated within the three inferred clusters, respectively.
>
> ### Table A. Average Score
>
>
> | Cluster  | ASUS VS197T-P 18.5 (B00B2HH7GK) | BLU R1 HD ArmorFlex Case + Screen Protector – White/Gold (B01GIRTG7G) | Kingston DT-Micro USB Flash 32GB, NeroB009CMN3V0 |
> |----------|----------------------------------|------------------------------------------------------------------------|---------------------------------------------------|
> | cluster1 | 0.5325                           | -0.0028                                                                | 0.0312                                            |
> | cluster2 | 0.0156                           | 0.2922                                                                 | 0.0109                                            |
> | cluster3 | 0.0429                           | 0.0281                                                                 | 0.0873                                            |
>
> ---
>
> ### Table B. Proportion
>
> | Cluster  | ASUS VS197T-P 18.5 (B00B2HH7GK) | BLU R1 HD ArmorFlex Case + Screen Protector – White/Gold (B01GIRTG7G) | Kingston DT-Micro USB Flash 32GB, NeroB009CMN3V0 |
> |----------|----------------------------------|------------------------------------------------------------------------|---------------------------------------------------|
> | cluster1 | 78.50%                           | 0                                                                      | 0                                                 |
> | cluster2 | 0                                | 39.55%                                                                 | 0                                                 |
> | cluster3 | 0                                | 0                                                                      | 7.57%                                             |
>
> The tables demonstrate that PRUC effectively identifies meaningful user clusters, with each cluster exhibiting a distinct preference for one of the three electronic products.
>
> We will include the above discussion in the camera-ready version of the paper.

---

> ### Author Response · Authors · 2024-12-02
> **Sincerely Looking Forward to Further Discussion**
>
> Dear Reviewer ny3m,
>
> Thank you for your review and engagement during the discussion period.
>
> In response to your suggestions, we have:
>
> - Revised the clarity and conciseness of the PDF to improve readability.
>
> - Conducted a comprehensive theoretical analysis of the clustering method, focusing on its motivation, complexity, scalability, and role.
>
> - Performed additional experiments, as shown in **(Rebuttal) Table A** and **(Rebuttal) Table B**, which demonstrate that different clusters exhibit preferences for different items.
>
> With the ICLR Discussion period concluding soon (Dec. 2nd (AOE) for reviewers and Dec.3rd (AOE) for authors), we kindly request your feedback on whether our responses address your concerns or if there are additional questions or suggestions you would like us to address.
>
> Thank you once again for your time!
>
> Best,
>
> The Authors

---

### Meta-Review · Area_Chair_7XGa · 2024-12-21

**Metareview:**

This paper introduces Probabilistic Residual User Clustering (PRUC), a method designed to address the complexity and lack of transparency in modern deep learning-based recommender systems. These systems often pose challenges in understanding and optimizing recommendations. PRUC tackles this by automatically clustering users, uncovering causal relationships among latent variables, and creating specialized models for each cluster using causal reasoning. By leveraging do-calculus to integrate rating residuals influenced by causal factors, PRUC significantly improves recommendation quality. Experimental results show that PRUC can be seamlessly integrated into various deep-learning recommender systems, yielding performance gains and enabling the automated identification of meaningful user clusters.

The reviewers propose the following strengths of the paper:
- The proposed idea is interesting and can solve real-world problems
- Extensive experiments are conducted to demonstrate the effectiveness of the proposed model

However, many negative points are also proposed by the reviewers:
- The paper writing can be further improved.
- The motivation for employing confounders can be further clarified.
- Lacks comparison with previous state-of-the-art approaches.

**Additional Comments On Reviewer Discussion:**

In the rebuttal period, the authors have addressed many concerns of the reviewers. However, some concerns can not be easily alleviated (such as the motivation of the paper). Considering that all the reviewers give negative scores on the paper, I tend to reject the paper.

---

### Decision · Program_Chairs · 2025-01-22

Reject